# Transcriptomics Integrated with Metabolomics: Assessing the Central Metabolism of Different Cells after Cell Differentiation in *Aureobasidium pullulans* NG

**DOI:** 10.3390/jof8080882

**Published:** 2022-08-22

**Authors:** Nan Zeng, Ning Zhang, Xin Ma, Yunjiao Wang, Yating Zhang, Dandan Wang, Fangxiong Pu, Bingxue Li

**Affiliations:** 1College of Bioscience and Biotechnology, Shenyang Agricultural University, Shenyang 110866, China; 2College of Land and Environment, Shenyang Agricultural University, Shenyang 110866, China

**Keywords:** stress defense, ROS, *Aureobasidium pullulans*, SCs, YL, transcriptomics, metabolomics, glyoxylate cycle

## Abstract

When organisms are stimulated by external stresses, oxidative stress is induced, resulting in the production of large amounts of reactive oxygen species (ROS) that inhibit cell growth and accelerate cellular aging until death. Understanding the molecular mechanisms of abiotic stress is important to enhance cellular resistance, and *Aureobasidium pullulans*, a highly resistant yeast-like fungus, can use cellular differentiation to resist environmental stress. Here, swollen cells (SCs) from two different differentiation periods in *Aureobasidium pullulans* NG showed significantly higher antioxidant capacity and stress defense capacity than yeast-like cells (YL). The transcriptome and the metabolome of both cells were analyzed, and the results showed that amino acid metabolism, carbohydrate metabolism, and lipid metabolism were significantly enriched in SCs. Glyoxylate metabolism was significantly upregulated in carbohydrate metabolism, replacing the metabolic hub of the citric acid (TCA) cycle, helping to coordinate multiple metabolic pathways and playing an important role in the resistance of *Aureobasidium pullulans* NG to environmental stress. Finally, we obtained 10 key genes and two key metabolites in SCs, which provide valuable clues for subsequent validation. In conclusion, these results provide valuable information for assessing central metabolism-mediating oxidative stress in *Aureobasidium pullulans* NG, and also provide new ideas for exploring the pathways of eukaryotic resistance to abiotic stress.

## 1. Introduction

Microbial growth often faces a complex and variable environment, and it is crucial for microorganisms to maintain the osmotic pressure balance inside and outside the cell to prevent the hazards caused by osmotic pressure imbalance [1]. The production of ROS is one of the fundamental characteristics of oxygen metabolism across the tree of life, and when cells are stimulated by external stresses, an oxidative stress response occurs, resulting in the production of large amounts of ROS [2]. There are three main types of ROS present in cells [3]: the reduction of molecular oxygen (O_2_) generates superoxide anions (O^2−^), which are converted to hydrogen peroxide (H_2_O_2_) by superoxide dismutase and completely reduced to water or partially reduced to hydroxyl radicals (HO^–^) by peroxidase or glutathione peroxidase [4]. Excess ROS in cells can affect normal cellular functions, alter the ratio of unsaturated fatty acids to proteins [5], interfere with cell membrane transport functions [6], and lead to cell death [7]. 

Organisms have developed several specific mechanisms to protect themselves from damage by different ROS and, thus, enhance their resilience [8]. There is growing evidence in some bacterial cells that the glyoxylate cycle plays a very important role in the response to oxidative stress. In addition, some fungi and yeasts can further increase cellular resistance to adapt to their environment by means of cell differentiation [9,10,11,12]. 

The glyoxylate cycle is a two-step metabolic pathway that replaces the TCA cycle, bypasses the step in the TCA cycle that produces carbon dioxide, and is an anabolic pathway present in most protozoa, plants, bacteria, and fungi [13]. The glyoxylate cycle has been found to play a very important role in stress defense and pathogenesis in oxidative stress and antibiotic stress experiments on *Pseudomonas aeruginosa* and *Mycobacterium tuberculosis* [14,15]. However, the role of the glyoxylate cycle in stress defense in fungal cells is still little reported. 

*Aureobasidium pullulans* is a class of yeast that is widely distributed in nature and is extremely resistant to a wide range of extreme environments [16,17,18,19]. In addition, yeast cells are widely used in abiotic stress studies because of their fast growth rate, ease of cultivation, and lower cost than plants [20], making *Aureobasidium pullulans* an ideal biological material for studying cellular resilience. Previous studies have shown that *Aureobasidium pullulans* has a unique cellular polymorphism [21], which can produce different morphologies in response to changes in the environment, and its lifecycle is mainly YL (pH ≥ 6.0) at the early stage, but they gradually differentiate to form SCs (pH = 4.5) to adapt to changes in the living environment with changes in environmental pH, and SCs as the central part of its lifecycle can produce a variety of important metabolites such as pullulans and polymalic acid [22,23]. At the end of its lifecycle, SCs can also form septate swollen cells (SSCs), chlamydospores (CHs), mycelium, etc. [24]. However, the mechanism of cell differentiation for resistance in yeast needs to be further investigated.

Therefore, the aim of this study was to analyze the central metabolic profile of *Aureobasidium pullulans* NG and the potential differences between SCs and YL after cell differentiation by integrating the transcriptome and the metabolome of the *Aureobasidium pullulans* NG grown in different cellular forms. The antioxidant capacity and growth of the two types of cells in response to external stresses were examined, followed by analysis of the transcriptomic and metabolomic data to further identify the potential differential accumulation of metabolites and the corresponding differentially expressed genes at the biochemical and molecular biology level. The results of this study may provide new insights into the stress defense mechanisms of eukaryotes.

## 2. Materials and Methods

### 2.1. Microorganism and Cultivation Conditions

*Aureobasidium pullulans* NG (CGMCC 3.12877) was inoculated into a YND medium (yeast extract, 0.02%; dextrose, 2%; NaNO_3_, 0.64%; MgSO_4_·7H_2_O, 0.05%; KH_2_PO_4_, 0.5%; pH 6.0) with shaking (180 rpm) at 28 °C for 24 h on a rotary shaker to obtain a pre-inoculum. One percent (*v*/*v*) pre-inoculum was transferred to a fresh YND medium with shaking (200 rpm) at 28 °C for 24 h to obtain an inoculum (OD_560nm_, 10.0). Then, 1% (*v*/*v*) pre-inoculum was transferred to a fresh YPD medium (10 g/L yeast extract, 20 g/L peptone, and 20 g/L glucose), and the cells were collected after shaking (200 rpm) at 28 °C for 48 h.

### 2.2. Sample Collection and Observation

Cells were collected by adding an appropriate amount of an *Aureobasidium pullulans* NG culture solution to sterile EPS tubes (2 mL) at 5000 rpm for 8 min. A 50 mmol/L Tris HCl buffer (pH 7.5) was prepared to rinse the organisms, followed by 1 mL buffer to mix them by blowing and prepare a cell suspension. One hundred percent Percoll isotonic stock solution (Beijing Solabao Technology Co., Ltd., Beijing, China) was diluted to 70% with 50 mmol/L (pH 7.5) Tris HCl buffer as an isolation medium. Two milliliters of the cell suspension were taken in a sterile EPS tube at 2000 g for 15 min. Four hundred microliters of the cell suspension were spread evenly and slowly over the centrifuged cell suspension, set at 5000 rpm for 30 min; the upper layer was YL and the lower layer was SCs.

### 2.3. Electron Microscopic Observation of SCs and YL

We improved the method of Murata et al. by using transmission electron microscopy to observe the two types of cells sorted out [25]. The collected cells were prepared as a suspension of an appropriate concentration, and 50 μL of the suspension were subsequently added dropwise to a copper grid carbon-coated with a formvar film; the sample was dried and negatively stained with 2% phosphotungstic acid solution under light-proof conditions and then observed with transmission electron microscopy (TEM, HT7700, HITACHI, Tokyo, Japan) under an accelerating voltage of 100 kV.

### 2.4. Treatment of the Two Types of Cells under Different Conditions

The selected SCs and YL were washed three times with a PBS buffer, and equal amounts of cells were placed in the PBS buffer, to which 0% (control), 1%, 5%, and 10% NaCl were added as salt stress treatment; 0% (control), 10%, 15%, and 20% H_2_O_2_ solutions were added as hydrogen peroxide stress treatment; pH was adjusted to 4.4, 6.0 (control), and 9.0, respectively, as acid–base stress treatment. All of the above stress treatments were incubated at 28 °C for 24 h in a shaker at 180 rpm. The temperature was set at 4 °C, 28 °C (control), and 32 °C as the temperature stress, and the cells were sampled after 24 h of incubation at 180 rpm. Cell mortality was measured using Melan staining and a hematocrit plate.

### 2.5. Determination of Intracellular Reactive Oxygen Species

The sorted SCs and YL were placed in 2 mL centrifuge tubes and added to nine times the volume of the phosphate buffer (PBS, pH 7.0). The fluorescence method was used to assay the relative levels of ROS using oxidant-sensitive probe 2′,7′-dichlorodihydrofluorescein diacetate (DCFH-DA) (Nanjing Jian-cheng Bio-engineering Institute, Nanjing, China) as described previously. The fluorescence intensity was detected using an F-4500 fluorescence spectrophotometer (Hitachi, Tokyo, Japan) with the settings of emission at 530 nm, of excitation—at 490 nm.

### 2.6. Determination of Intracellular Superoxide Dismutase, Catalase and Total Antioxidant Capacity

The sorted SCs and YL were placed in 2 mL centrifuge tubes, added to nine times the volume of the phosphate buffer solution (PBS, pH 7.0), and then sonicated under ice water bath conditions to make 10% tissue homogenate, centrifuged for 10 min at 2500 rpm, and the supernatant was transferred to a new EP tube. The water-soluble tetrazolium salt (WST-1) method was used to assay relative activities of superoxide dismutase (SOD) using a Super-oxide Dismutase Assay Kit (Nanjing Jiancheng Bio-engineering Institute, Nanjing, China) according to operation specifications. The ammonium molybdate method was used to assay relative activities of catalase (CAT) using a catalase assay kit (Nanjing Jiancheng Bio-engineering Institute, Nanjing, China) according to operation specifications. Total antioxidant capacity (T-AOC) was determined using a Total Antioxidant Capacity Kit (Beijing Solabao Technology Co., Ltd., Beijing, China). SOD and CAT activities and T-AOC were evaluated using a Multiskan FC enzyme marker (Thermo, Waltham, MA, USA) with optical density values set to 450, 405, and 593 nm, respectively.

### 2.7. Transcriptome Analysis

Total RNA extraction was performed using the Trizon reagent. The extracted RNA was assessed for integrity using an Agilent 2100 Bioanalyzer. The transcriptome sequencing analysis was compared using the next-generation sequencing (NGS) technology based on the Illumina HiSeq sequencing platform for SCs and YL. The obtained high-quality sequences were spliced from scratch to obtain transcript sequences, which were then clustered and analyzed to select the longest transcripts as UniGene, and finally UniGene was used for subsequent analysis; the filtered sequences were also compared to UniGene, and then the samples were further analyzed for expression differences and enrichment. Gene expression levels were estimated from unique mapping reads using the reads per kilobase million mapping (RPKM) method of RSEM v1.2.15. Hierarchical clustering analysis was performed according to the Pearson correlation and mean linkage methods. GO enrichment analysis was performed using the Blast2GO (v2.5) software. Kegg enrichment analysis was performed using the KOBAS (v3.0) software. Differential gene expression was performed using the DESeq (v1.32.0) program with default parameters. These expressed genes with log_2_ fold change ≥1 and adjusted *p*-value (padj) ≤ 0.05 were considered differentially expressed genes (DEGs). The pathway analysis of DEGs was performed using the KEGG PATHWAY database (http://www.kegg.jp (accessed on 3 August 2020)) and Blast2GO.

### 2.8. Metabolome Analysis

Metabolite extraction was performed using the method of Sangster et al. [26] with Smart et al. [27]. Chromatographic separation was accomplished in a Thermo Vanquish system equipped with an ACQUITY UPLC^®^ HSS T3 (150 × 2.1 mm, 1.8 µm, Waters) column maintained at 40 °C. The temperature of the autosampler was 8 °C. Gradient elution of analytes was carried out with 0.1% formic acid in water (B2) and 0.1% formic acid in acetonitrile (A2) or 5 mM ammonium formate in water (B1) and acetonitrile (A1) at a flow rate of 0.25 mL/min. Injection of 2 μL of each sample was performed after equilibration. An increasing linear gradient of solvent A (*v*/*v*) was used as follows: 0~1 min, 2% A2/A1; 1~9 min, 2~50% A2/A1; 9~12 min, 50~98% A2/A1; 12~13.5 min, 98% A2/A1; 13.5~14 min, 98~2% A2/A1; 14~20 min, 2% A2-positive model (14~17 min, 2% A1-negative model).

ESI–MSn experiments were executed on a Thermo Q Exactive Focus mass spectrometer with the spray voltage of 3.8 kV and −2.5 kV in positive and negative modes, respectively. Sheath gas and auxiliary gas were set at 30 and 10 arbitrary units, respectively. The capillary temperature was 325 °C. The analyzer scanned over a mass range of m/z 81–1000 for a full scan at a mass resolution of 70,000. Data-dependent acquisition (DDA) MS/MS experiments were performed with HCD scanning. The normalized collision energy was 30 eV. Dynamic exclusion was implemented to remove some unnecessary information in the MS/MS spectra. Metabolite identification was based on the HMDB (http://www.hmdb.ca (accessed on 3 August 2020)), Metlin (http://metlin.scripps.edu (accessed on 3 August 2020)), MassBank (http://www.massbank.jp (accessed on 3 August 2020)), LipidMaps (http://www.lipidmaps.org (accessed on 3 August 2020)), and mzClound (https://www.mzcloud.org (accessed on 3 August 2020)) databases.

### 2.9. Real-Time Quantitative PCR Confirmation of the RNA-Seq Data

The transcript levels of target genes were detected using real-time quantitative PCR (qPCR) performed using a ChamQ Universal SYBR qPCR Master Mix kit (Vazyme Biotech Co., Ltd., Nanjing, China) in a Quant Studio6 Flex system (Thermo Fisher Scientific, Waltham, MA, USA). As the internal reference gene, 18SrDNA was selected. To analyze the qPCR results, the relative expression of each gene was calculated by the comparative crossing point (CP) method and expressed as 2^−∆∆CT^. Each gene expression analysis was performed using three independent biological replicates. The specific primers used for qPCR are listed in Appendix A.

### 2.10. Statistical Analysis

The qPCR data of the selected gene expression profiles were analyzed using software package SPSS 19.0 for Windows (SPSS Inc., Chicago, IL, USA). Prior to analysis, the data were normalized by transforming as ln (x + 1). All the data were expressed as the means ± standard error of at least triplicates of independent cultures. The means of the different treatments were compared using one-way ANOVA using Tukey’s honestly significant difference test at the 5% probability level (*p* < 0.05).

## 3. Results

### 3.1. Determination of the Antioxidant Capacity of SCs and YL

Two types of cells, SCs and YL, were collected by Percoll density gradient centrifugation and observed by electron microscopy, and it was found that the SCs had an obvious film coating outside (Figure 1A), while the YL mainly showed a long shuttle shape without a film coating (Figure 1B). Subsequently, CAT, SOD, T-AOC, and the respective ROS levels were measured for the SCs and YL to obtain the antioxidant capacity of both cells. As shown in Table 1, the levels of CAT, SOD, and T-AOC in the SCs were significantly higher than those in YL, with 7.64 ± 2.44 U/mg prot, 103.54 ± 7.71 U/mg prot, and 7.949 ± 0.017 U/mg prot, respectively. Moreover, detection of the ROS levels in the SCs and YL showed that the levels of ROS in the SCs were significantly lower than those in YL (Figure 2). All these results indicated that the antioxidant capacity of the SCs was higher than that of YL.

### 3.2. Survival of Two Types of Cells in Response to Different External Stress Conditions

Microorganisms under environmental stress usually cause oxidative stress and, thus, generate excess ROS to affect cell growth. We had previously verified that the antioxidant capacity of SCs is significantly higher than that of YL, and then we examined the growth of both types of cells cultured under different adversity conditions.

Experiments showed that the mortality rates of both the SCs and YL showed an increasing trend with increasing salt concentration under NaCl stress (Figure 3A). When the salt concentration was 1%, there was no significant difference between the mortality rates of the YL and SCs, and when the salt concentration increased to 5%, the mortality rate of the SCs was 29%, which was significantly lower than that of the YL (54%), which indicated that the SCs possessed the property of resistance to the adversity stress when the salt concentration increased to a certain degree; when the salt concentration increased to 10%, the mortality rate of the SCs was significantly lower than that of the YL, and the mortality rates of the two forms of cells were 34% and 74%, respectively. Under no exogenous H_2_O_2_ stress, there was no significant difference in mortality between the YL and SCs (Figure 3B); after the stress treatment with the addition of 10% H_2_O_2_ solution, the difference in mortality between the two forms of cells was not significant; when the concentration of the H_2_O_2_ solution was increased to 15%, a significant difference appeared between the two forms of cells, and the mortality rate of the YL rose to 69%, while that of the SCs was 30%; when the H_2_O_2_ solution was continuously increased to 20%, the mortality rate of the YL reached 79%, while that of the SCs was 36%. When the pH of the medium was 6.0, which was the optimal culture condition for *Aureobasidium pullulans* NG, both the YL and SCs could grow and reproduce normally (Figure 3C); when the pH was 4.4, the YL showed a higher mortality rate compared to the SCs, up to 71%; when the pH was increased to 9.0, the mortality rates of the YL and SCs were 74% and 23%, respectively. At the optimum temperature of 28 °C, there was little difference in mortality between the SCs and YL, and both could grow and reproduce normally at this temperature (Figure 3D); when the SCs and YL were subjected to low-temperature stress at 4 °C, it could be seen that the YL could not resist the low-temperature stress and had a mortality rate of 65%, while the mortality rate of the SCs was only 32%; when the two cell forms were subjected to high-temperature stress at 32 °C, it could be seen that the SCs had obvious resistance to high-temperature stress, while the mortality rate of the yeast-like cells was up to 77%. The combined experiments showed that the SCs had a stronger ability to resist environmental stress compared to the YL and were more adapted to environmental changes and, thus, to their increase.

### 3.3. Transcriptome Analysis

To identify differentially expressed genes (DEGs), six transcriptome cDNA libraries were constructed using next-generation sequencing (NGS) based on the Illumina HiSeq sequencing platform. By sequencing six cDNA libraries, 38.89 Gb of high-quality sequence bases were obtained. The conditions for screening differentially expressed genes were as follows: expression difference fold (log_2_ fold change) > 1, significance *p*-value < 0.05. A total of 5585 differentially expressed genes (DEGs) were screened compared with the control (YL), and there were 2798 genes with upregulated expression and 2787 genes with downregulated expression in the SCs (Figure 4A,B). These differentially expressed genes may play a key role in the mechanism of oxidative stress in response to SCs.

In order to further understand the biological functions of these differentially expressed genes and the related biological processes they were involved in, we performed correlation studies on GO and KEGG enrichment analyses, respectively. For GO analysis, DEGs of the SCs were enriched to 30 items in the GO categories of cell composition (CC), metabolic function (MF), and biological process (BP). As shown in Figure 4C, the top three terms significantly enriched in GO in the CC category were nuclear pore (GO:0005643), nuclear envelope (GO:0005635), and organelle envelope (GO:0031967). The top three GO terms significantly enriched in the MF category were structural constituent of nuclear pore (GO:0017056), substrate-specific transmembrane transporter activity (GO:0022891), and substrate-specific transporter activity (GO:0022892). In the BP category, the top three significantly enriched GO terms were nucleocytoplasmic transport (GO:0006913), nuclear transport (GO:0051169), and intracellular transport (GO:0046907). As shown in Figure 4D, KEGG analysis showed that a total of 115 pathways were enriched by DEGs in the SCs, and the top 20 significantly enriched pathways were mainly associated with metabolism, genetic information processing, cellular processes, and organismal systems. Thirteen enriched pathways were finally screened for their possible involvement in antioxidation in the SCs, including glyoxylate and dicarboxylic acid metabolism (ko00630), pyruvate metabolism (ko00620), fatty acid metabolism (ko00071), pentose phosphate pathway (ko00030), oxidative phosphorylation (ko00190), fructose and mannose metabolism (ko00051), starch and sucrose metabolism (ko00500), glycolysis and sugar metabolism (ko00010), alanine, aspartate, and glutamate metabolism (ko00250), tricarboxylic acid cycle (ko00020), glycerophospholipid metabolism (ko00564), vitamin B6 metabolism (ko00750), pantothenate and coenzyme A biosynthesis (ko00770), and, finally, 35 genes were screened for significant upregulation associated with SC antioxidants (Appendix A).

### 3.4. Metabolome Analysis

The metabolomic profiles were analyzed using the LC-MS/MS coupling technique. The results showed that 4821 upregulated metabolites and 1575 downregulated metabolites were detected in the positive ion mode in the treated group (SCs) and the control group (YL), and 2941 upregulated metabolites and 1219 downregulated metabolites were detected in the negative ion mode. The differences of metabolites in the SC and YL cells were then analyzed by using a PLS-DA model and an OPLS-DA model to investigate the metabolic changes and the response of the antioxidant capacity of different cells. As shown in Figure 5A,B, both the PLS-DA and OPLS-DA models showed good separation of each sample pair (SC vs. YL) (*p* < 0.05), and the alignment test indicated that the method had good model validity and accuracy. Then, the OPLS-DA model was used to screen out the differentially significant metabolites with the VIP threshold > 1 and *t*-test (*p* < 0.05), and finally a total of 173 differential metabolites were obtained in the SCs vs. YL, among which there were 52 upregulated metabolites and 121 downregulated metabolites (Figure 5C). KEGG enrichment analysis was performed to better understand the biological functions of these differential metabolites and the related biological processes they are involved in. The results showed that the differential metabolites between the SCs and YL were enriched in a total of 39 KEGG pathways (Appendix A). The first 20 KEGGs were highly consistent with the transcriptome and were mainly categorized as amino acid metabolism and carbohydrate metabolism.

To obtain more information on the physiological defenses such as resistance to adversity of different cells after cell differentiation of *Aureobasidium pullulans* NG, KEGG enrichment analysis was performed on the differentially expressed genes and metabolites with *p* < 0.05 in the transcriptome and metabolome results of the SCs and YL. Most of the 137 enrichment pathways obtained belonged to carbohydrate metabolism. Among these carbohydrate metabolism pathways, glyoxylate and dicarboxylate metabolism, starch and sucrose metabolism, pentose phosphate pathway, amino sugar and nucleotide sugar metabolism, pyruvate metabolism, glycolysis/gluconeogenesis, TCA cycle were significantly enriched, and most of the carbohydrates in these pathways were upregulated. Finally, we screened the 10 key genes and the two key metabolites related to antioxidants in the SCs (Table 2), which were all significantly upregulated, except for SC16410_0_3 and SC16898_5_17 which were downregulated (Figure 6).

## 4. Discussion

In this study, to investigate the differences in the performance of different cells against adversity and how *Aureobasidium pullulans* NG uses cell differentiation to resist stress, we tested the antioxidant capacity of the SCs and YL and determined the survival of both cells under different external stress conditions. These results showed that the antioxidant capacity and resistance to external stress were significantly higher in the SCs than in the YL. In addition, the main life forms in the lifecycle of *Aureobasidium pullulans* NG were the YL and SCs, which were mainly grown as YL (pH ≥ 6.0) in its early stage; however, YL gradually differentiated to form SCs (pH = 4.5) with the change of environmental conditions and the decrease in the nutrient level. The environmental condition of pH 4.5 is an adverse condition for YL, where the growth and development of YL is severely inhibited, yet SCs can adapt to this environment and grow and survive under this adverse condition. Therefore, we performed a transcriptome–metabolome analysis to further explore the differences between the two types of cells. The results showed that the amino acid metabolic pathway, carbohydrate metabolic pathway, and lipid metabolic pathway were significantly enriched in the SCs, and, finally, 10 genes and two metabolites were screened, which may be a key part of the two types of cells showing different abilities to resist external stress and also play an indirect regulatory role in the stress defense of *Aureobasidium pullulans* NG.

Previous studies showed that amino acid metabolism and carbohydrate metabolism play an important role in the resistance of eukaryotic cells to external stresses [28]. ROS are produced when cells are exposed to external stresses causing oxidative stress [28], and free amino acids are thought to play a key role as a regulator [29]. It was found that glutathione (GSH), a tripeptide containing cysteine, glutamate, and glycine residues, has multiple functions [30] and plays a crucial role in the regulation of ROS in living organisms [31]. Notably, the results of transcriptome and metabolome penetrance analysis revealed that cysteine, glutamate, and glycine metabolism were significantly enriched in the SCs and the related genes were significantly upregulated, while the upregulation of cysteine, glutamate, and glycine was mainly used for the synthesis of glutathione in cells and, thus, for the scavenging of ROS in vivo. In addition, glutathione S-transferase (GST) has been demonstrated as an effective antidote for the excess ROS in plants [32], and transcriptional analysis showed a 5.78-fold upregulation and significant accumulation in metabolites, resulting in the synthesis of glutathione to enhance cellular antioxidant properties.

As an important pathway for cell growth and energy metabolism, carbohydrate metabolic pathways are also involved in enhancing the ability of cells to resist external stresses [33]. Compared to the YL, most of the carbohydrates in the SCs were downregulated, while myo-inositol was significantly upregulated. Resistance to external stress through inositol upregulation in plant cells has been reported [34], as inositol acts as a precursor substance for a variety of important metabolites involved in a variety of physiological and biochemical processes, such as growth regulation, cell membrane formation, hormone regulation, programmed cell death, stress signaling, and cellular immunity, among other metabolic pathways [35]. In addition, it is noteworthy that the key genes in the glyoxylate cycle and dicarboxylic acid metabolic pathways were significantly upregulated in the carbohydrate metabolic pathway enriched by the SCs, while the key genes in the TCA cycle pathway were significantly downregulated. It has been shown that the glyoxylate cycle plays an important role in stress defense and resistance to adversity. In some bacteria, the glyoxylate cycle is activated as the main metabolic pathway to replace the TCA cycle when they are exposed to external stimuli and stresses and activates the related enzymes and their genes to assist in the removal of excess ROS produced in the cells during stress to maintain cell survival and adapt to the environment [14,15,36]. Moreover, transcriptional metabolomics penetration analysis revealed a significant enrichment of citric and succinic acid content in the SCs. Previous studies showed that citric acid significantly enhances the tolerance of plant cells to abiotic stresses and that citric acid may also regulate human osteoblast differentiation [37,38,39]. It is worth mentioning that in our recent study, it was found that the addition of citric acid could help *Aureobasidium pullulans* NG to maintain SC cell morphology and could significantly help it to resist salt stress (unpublished). Succinate is formed mainly by oxidation of stored fatty acids through the glyoxylate cycle, and its role is to provide energy to cells by synthesizing carbohydrates through the gluconeogenic pathway on the one hand and to provide substrates for other metabolic pathways to accelerate the rate of cellular metabolism to scavenge the ROS produced by environmental stress on the other.

Furthermore, the relationship between lipid metabolic pathways and biological resistance to adversity has been demonstrated [40,41]. In the transcriptome, the glycerolipid metabolic pathways are significantly enriched in SCs, and glycerol can play a very critical role in the resistance of cells to external stresses. External stress usually leads to the conversion of intracellular carbohydrate storage resources to lipids, thereby increasing total lipid biosynthesis to eliminate excess ROS [42,43]. Thus, the synthesized glycerol and other related lipid metabolites are mainly used to provide substrates for increased amino acids and carbohydrates as well as to participate in energy metabolism to provide energy, enhancing the antioxidant capacity of cells in an indirect way.

The glyoxylate cycle has been considered as a complementary pathway to the TCA cycle [44], but there is growing evidence that when some prokaryotic cells are subjected to external stimuli, the glyoxylate cycle not only replaces the central metabolic hub of the TCA cycle to provide energy to cells, but also upregulates the glyoxylate cycle and activates the related enzymes and genes to help them resist adversity [13]. However, the pathway by which the glyoxylate cycle regulates ROS remains unclear, and this role of the glyoxylate cycle in helping cells resist adversity has not been identified in yeast cells. Therefore, we constructed a mechanistic map of the stress defense response in SCs (Figure 7A) and YL (Figure 7B) of *Aureobasidium pullulans* NG. When SCs are subjected to external stimuli, the key genes in the TCA cycle are downregulated, while the glyoxylate cycle is upregulated to replace the central hub of cellular metabolism in the TCA cycle, which provides substrates and energy for amino acid metabolism, carbohydrate metabolism, and lipid metabolism to maintain cell survival and accelerate the cellular metabolic rate to clear the ROS generated by stress; on the other hand, upregulation of the glyoxylate cycle activates antioxidant-related enzymes and the key genes, which further helps cells to scavenge ROS and enhance their antioxidant capacity, thus helping them to adapt to the environment and resist adversity. However, when YL are subjected to external stress, the enzymes and genes related to antioxidative stress in the cell are not mobilized to help it scavenge ROS, and eventually ROS accumulate and cause cell death.

## 5. Conclusions

In conclusion, we tested the antioxidant capacity and stress defense capacity of SCs and YL in *Aureobasidium pullulans* NG and performed transcriptome–metabolome analysis of both types of cells to investigate the differences in stress defense exhibited by the two types of cells and the changes in central metabolism of the two types of cells after cell differentiation in *Aureobasidium pullulans* NG. We found that the antioxidant capacity and stress defense of the SCs were significantly higher than those of the YL. In addition, we successfully identified the 10 key genes and the two key metabolites involving amino acid metabolism, carbohydrate metabolism, and lipid metabolism. In the analysis of the transcriptome and the metabolome, we found that glyoxylate metabolism plays a key role in the stress defense of SCs, which also demonstrates that *Aureobasidium pullulans* NG can use cell differentiation to help it adapt to its environment against adversity. This study lays the foundation for elucidating the mechanism of resistance of *Aureobasidium pullulans* NG to external stresses and also provides molecular clues for yeast cells to resist abiotic stresses.

## Figures and Tables

**Figure 1 jof-08-00882-f001:**
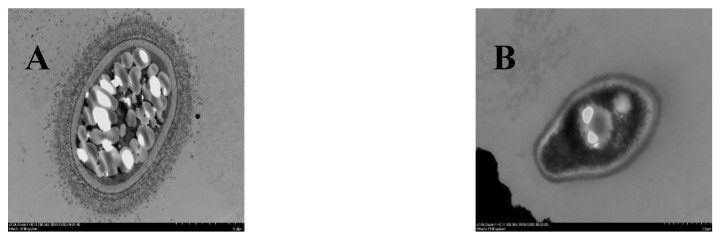
Transmission electron microscopy of two kinds of cells. (**A**) SCs, bars = 5 μm; (**B**) YL, bars = 1 μm.

**Figure 2 jof-08-00882-f002:**
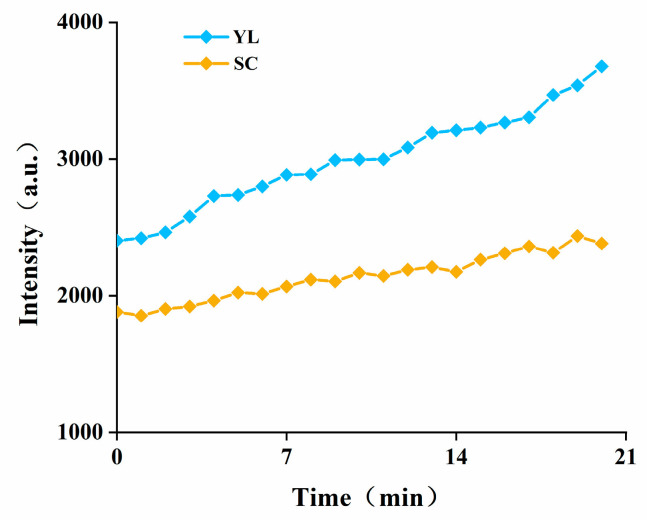
Comparison of the intracellular ROS levels between the SCs and YL in *Aureobasidium pullulans NG.* The data represent the means ± standard deviations of triplicate experiments.

**Figure 3 jof-08-00882-f003:**
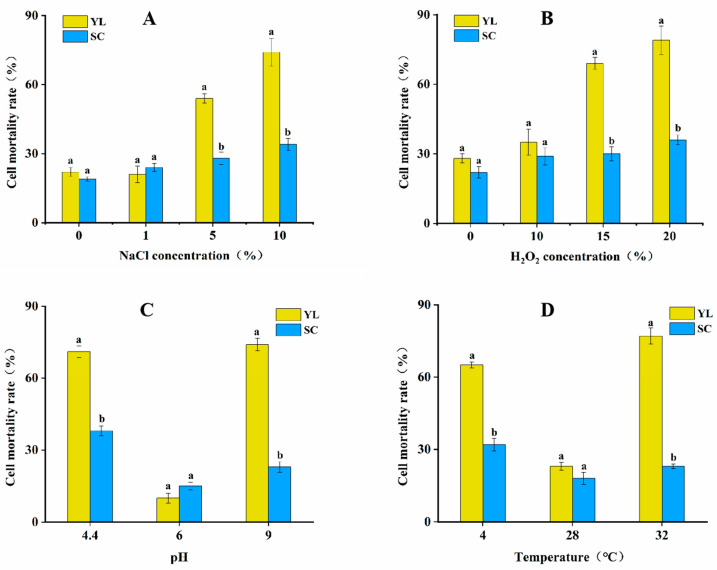
Survival of two types of cells in response to different external stress conditions. (**A**) Effect of NaCl treatment on the mortality of the two types of cells. (**B**) Effect of H_2_O_2_ treatment on the mortality of the two types of cells. (**C**) Effect of pH change on the mortality of the two types of cells. (**D**) Effect of temperature on the mortality rate of both types of cells. Different letters represent significant differences between sample treatments (*p* < 0.05).

**Figure 4 jof-08-00882-f004:**
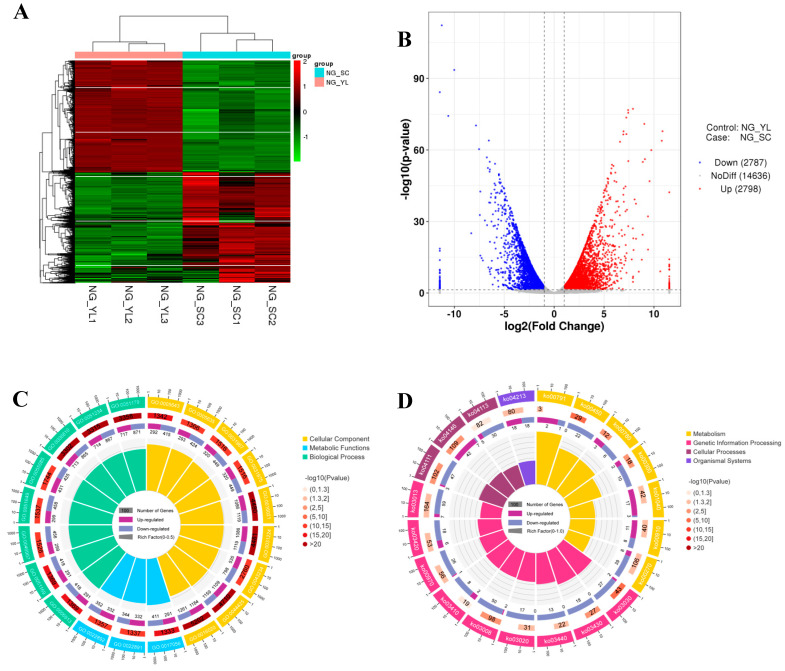
Transcriptome analysis of the SCs and YL in *Aureobasidium pullulans* NG. (**A**) Clustering analysis of the differentially expressed genes of the SCs and YL. (**B**) Volcano plot of the differentially expressed genes of the SCs and YL. (**C**) Circos plots of GO enrichment analysis of the DEGs. (**D**) Circos plots of KEGG enrichment analysis of the DEGs. From the outer circle to the inner, the top 20 enriched GO terms or KEGG pathways (ring 1), numbers of background genes in the genome (ring 2), numbers of upregulated and downregulated genes (ring 3), and rich factor of the DEGs in the corresponding GO terms or KEGG pathways (ring 4) are represented.

**Figure 5 jof-08-00882-f005:**
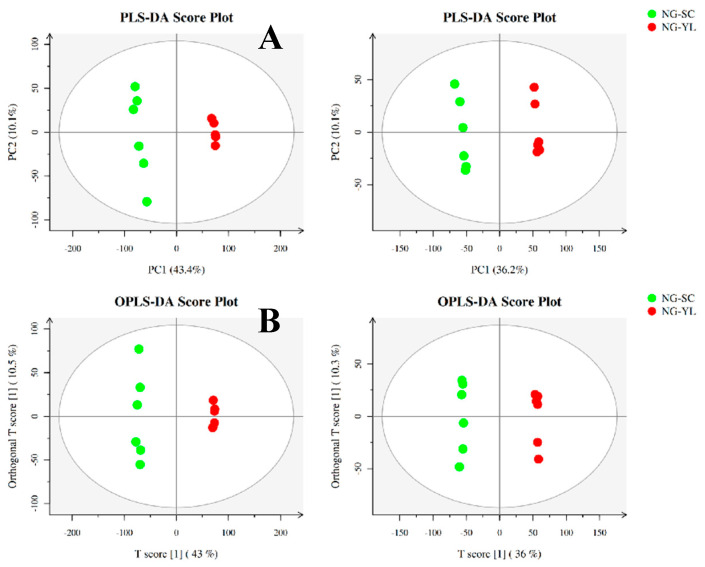
Metabolome analysis of the SCs and YL in *Aureobasidium pullulans* NG. (**A**) PLS−DA plots between the treated group (SCs) and the control group (YL). (**B**) OPLS−DA plots between the treated group (SCs) and the control group (YL). POS and NEG represent the positive and negative ion modes of mass spectrometry, respectively. (**C**) Hierarchical cluster and heatmap displaying the abundances levels of DAMs in all the samples.

**Figure 6 jof-08-00882-f006:**
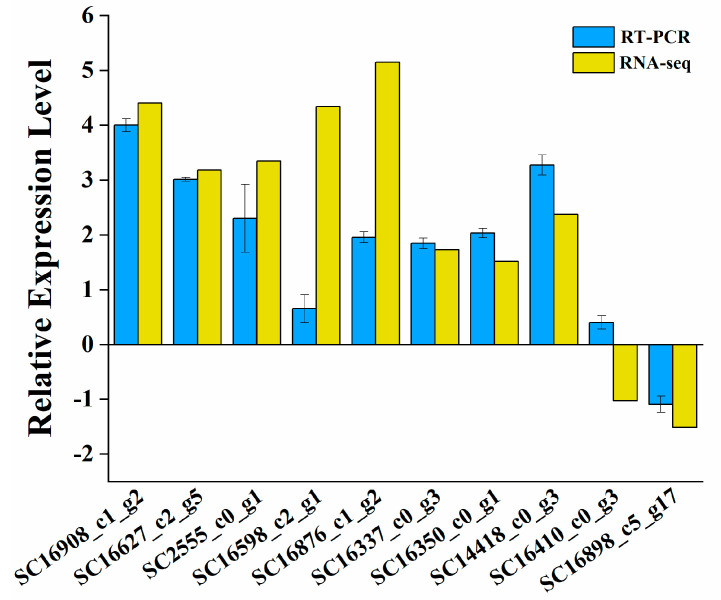
Real−time quantitative PCR detection of the key resistance regulatory genes between the SCs and YL in *Aureobasidium pullulans* NG. At least three independent experiments were performed in all the experiments. The data represent the average of biological replicates with standard deviation.

**Figure 7 jof-08-00882-f007:**
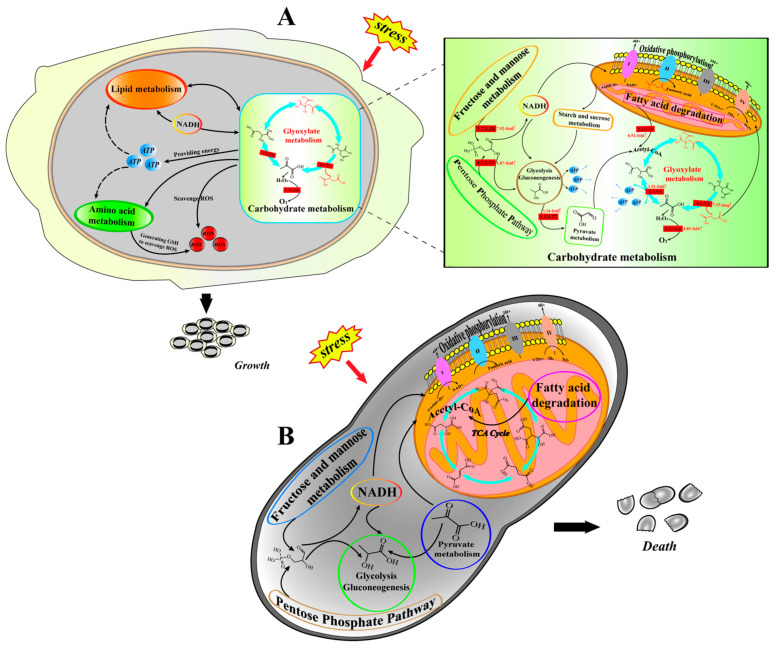
Mechanic maps of stress resistance in SCs (**A**) and YL (**B**) in *Aureobasidium pullulans* NG. The red chemical formulas in the graph are the key antioxidant metabolites, citric acid, and succinic acid, respectively. The red squares are upregulated genes followed by the upregulation multiplier.

**Table 1 jof-08-00882-t001:** Determination of the antioxidant capacity of the SCs and YL. Values are the means ± standard error of three independent replicates.

	Catalase (CAT) Activity (U/mg)	Superoxide Dismutase (SOD) Activity (U/mg)	SOD Inhibition Rate (%)	Total Antioxidant Capacity (T−AOC) (U/mg)
Yeast cells	2.87 ± 0.74 ^b^	63.03 ± 1.60 ^b^	60	4.067 ± 0.011 ^b^
Swollen cells	7.64 ± 2.44 ^a^	103.54 ± 7.71 ^a^	69	7.949 ± 0.017 ^a^

Note: U (unit) means the amount of enzyme required to catalyze the reaction of the substrate per unit of time. Different letters represent the degree of significant difference between different samples (*p* < 0.05).

**Table 2 jof-08-00882-t002:** Key genes and metabolites for SC resistance to adversity from integration analysis of metabolomic and transcriptomic data.

Classification	ID	Description	Up/Downregulated
UniGene	SC16908_c1_g2	Catalase domain-containing protein	Up
	SC16627_c2_g5	CAT2 catalase	Up
	SC2555_c0_g1	L-lactate dehydrogenase	Up
	SC16598_c2_g1	Acyl-CoA oxidase	Up
	SC16876_c1_g2	Alcohol dehydrogenase	Up
	SC16337_c0_g3	Glyceraldehyde-3-phosphate Dehydrogenase	Up
	SC16350_c0_g1		Up
	SC14418_c0_g3	Alpha-amylase	Up
	SC16410_c0_g3	Citrate synthase	Down
	SC16898_c5_g17	Aconitate hydratase	Down
Metabolites	com00158	Citrate	Up
	com00042	Succinate	Up

## Data Availability

The data presented in this study are available upon request from the corresponding author. The RNA-seq data are not publicly available because other data from these transcriptomes and metabolomes are being used for other analyses to be published independently from this one.

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
