# Peer review of "Transcriptomics Integrated with Metabolomics: Assessing the Central Metabolism of Different Cells after Cell Differentiation in Aureobasidium pullulans NG"

_jof, 2022, doi:10.3390/jof8080882_

Round 1
Reviewer 1 Report
The authors investigate differences in ROS and antioxidants in swollen cells and yeast-like cells of Aureobasidium pullans. They carried out transcriptomic and metabolic analyses. They identify genes and metabolites that are upregulated in swollen cells relative to yeast-like cells. The authors propose a model to explain the antioxidant effects of e.g. glyoxylate cycle upregulation in swollen cells and the stress sensitivity of yeast-like cells.
The article is of wide interest and is well set-out. It does a good job of interpreting the results and setting them in the context of the literature. The article should be accepted with minor amendments. Mostly these are English language errors but two sections of the results have no accompanying figures or tables. This should be checked.
52/53 Pseudomonas aeruginosa should be in Italics
53 Mycobacterium tuberculosis should be in Italics
54 Aureobasidium pullans should be in Italics
56 cellsarewidely should be cells are widely
57 andlowercost should be and lower cost
58 Aureobasidium pullans should be in Italics
59 Aureobasidium pullans should be in Italics
62 graduallydifferentiates should be gradually differentiates
63 changesin should be changes in
65 SCcanalso should be SC can also
68 Aureobasidium pullans should be in Italics
90 Is "blowing" the correct word
94 r/min should be rpm
96/97 either say nine times the volume or say ratio of 1:9
104/105 either say nine times the volume or say ratio of 1:9
107 r/min should be rpm
137 an should be a
216 Figure 3C masks figure 3A and figure 3D masks figure 3B
271 Figure 4A and 4B missing
264-270 Where is figure or table to support statments?
277-286 Where is figure or table to support statments?
Author Response
We thank the reviewer for his time and evaluation of our manuscript.
Please, find enclosed our response.

Reviewer 2 Report
The main problem with this manuscript is that lacks clarity in terms of the pursued aims and experimental design.
The authors stated the following: "Therefore, the aim of this study was to discuss the effect of cell differentiation on yeast cell resilience in Aureobasidium pullulans and how the glyoxalate cycle helps yeast cells resist oxidatives tress, which achieved by comparing the antioxidant capacity of SC with YL in Aureobasidium pullulans NG and the metabolome with the transcriptome." First, the statement is confusing, as they tried to integrate cell differentiation, antioxidant activities, and glyoxylate cycle in one complex aim, which honestly, is hard to understand. Following the experimental design, it seems that the authors wanted to assess the transcriptional and metabolomic differences between swollen cells and yeast-like cells. If this is the aim, then the introduction section and discussion should be rewritten; if not, the experimental design is not the correct one to address the different abilities of response to oxidative stress of both cell morphologies. If this would be the case, then cells should be treated with oxidative compounds before the extraction of RNA or metabolites. The conclusion mentioned that "In conclusion, a comprehensive transcriptomic and metabolomic analysis was performed to investigate the central metabolic response of Aureobasidium pullulans NG to resistance to oxidative stress." This is not correct the authors did not include analyzes of all the transcriptomic and metabolic data, they only focused on the aspects of their interest; thus I do not think they are comprehensive analyses. Second, the authors did not set any experiment to assess resistance to oxidative stress. As mentioned, only compared two cell morphologies.
As minor issues:
The manuscript need proofreading by a professional editing service and the authors should use italics for the scientific names.
Some parts of the methodology are written like a laboratory manual.
The manuscript does not include the methodology to perform electron microscopy
The results lack statistical analyzes
What is life history?
In table 1, how are defined the Units?
Author Response

(The authors gave the same response as above.)

Round 2
Reviewer 2 Report
I could find a point-by-point reply to all my previous concerns. However, I read the revised version of the manuscript and did not see changes addressing my initial comments that still stand. Please address them and reply to them in a rebuttal letter.

Author Response
We thank the reviewer for his time and evaluation of our manuscript.
Please, find enclosed our response.
Response to Reviewer 2 Comments
Point 1: The authors revised the text and included the following statement: Therefore, the aim of this study was to discuss the ability of different cells in Aureobasidium pullulans NG to resist environmental stress after cell differentiation and the effect of cell differentiation on yeast cell resilience, achieved by comparing the antioxidant capacity of SC and YL in Aureobasidium pullulans NG, and their growth in response to external stress.
If this is the aim, once again the experimental design is wrong, the authors did not analyze cells under stress. If the aim is changed to only analyze the transcriptome and metabolome of Aureobasidium pullulans NG growing in different cell morphologies, then the aim is aligned with the experimental design.
Response 1: Thank you once again for taking your time to provide valuable comments on our manuscript. We have made changes to the manuscript taking into account your suggestions.
Point 2: please expand this aspect in a different subsection of materials and methods. Section 2.2 should be written in the past tense.
Response 2: Thank you for your valuable advice.we have changed section 2.2 of the manuscript to the past tense and expanded the electron microscopy method to a new section in manuscript 2.3.
Point 3: then, it should be life cycle.
Response 3: Thank you for your comment.We have changed the life history in the manuscript to life cycle (Full text correction).
Point 4: I am asking what it means U (unit). It should be a certain amount of product generated per time unit; this should be explained in the text.
Response 4: We appreciate you pointing this out, and we have added a note for U (unit) in Table 1 of the manuscript.
Round 3
Reviewer 2 Report
The manuscript is suitable for publication.